# The Adverse Effects of Tuberculosis Treatment: A Comprehensive Literature Review

**DOI:** 10.3390/medicina61050911

**Published:** 2025-05-17

**Authors:** Rūta Mereškevičienė, Edvardas Danila

**Affiliations:** Institute of Clinical Medicine, Clinic of Chest Diseases, Immunology and Allergology, Faculty of Medicine, Vilnius University, LT-03101 Vilnius, Lithuania; edvardas.danila@santa.lt

**Keywords:** tuberculosis, adverse drug reactions, first-line drugs, second-line drugs, drug-susceptible tuberculosis, drug-resistant tuberculosis

## Abstract

Tuberculosis remains a significant public health challenge globally. The emergence of multidrug-resistant Mycobacterium tuberculosis strains presents one of the biggest hurdles in tuberculosis management. Both first- and second-line tuberculosis drugs are associated with common adverse reactions, which can lead to treatment interruptions and decreased adherence. In this article, we review the most commonly used drugs for the treatment of tuberculosis, focusing on the adverse reactions they may cause. We will examine the frequency and timeline of adverse drug reactions involving gastrointestinal, cardiac, neurological, nephrological, and cutaneous systems. Identifying patients at risk of developing those reactions is crucial for healthcare providers to implement monitoring strategies and manage complications effectively. In the review, we present the data about risk factors, management recommendations, and drug discontinuation rates as a result of side effects.

## 1. Introduction

Tuberculosis remains a significant public health challenge on a global scale. In 2023, it was estimated that approximately 10.8 million individuals fell ill with tuberculosis, a disease that affects populations in all countries worldwide. Of the 10.8 million cases, 6 million were men, 3.6 million were women, and 1.3 million were children [1]. Tuberculosis has been reaffirmed as the leading cause of death attributed to a single infectious agent, with a staggering 1.25 million fatalities reported in 2023. Among these, 161,000 deaths occurred in individuals co-infected with HIV [1].

The main causative agent of tuberculosis is Mycobacterium tuberculosis. It is a rod-shaped obligate aerobe that can be inhaled into the lungs, where it inhabits and slowly multiplies within macrophages. Despite being an ancient disease, tuberculosis still poses difficulties for healthcare professionals who are treating it. First-line antituberculosis drugs that are used widely for drug-susceptible tuberculosis treatment are rifampicin, isoniazid, ethambutol, and pyrazinamide. Meanwhile, the second-line drugs are prescribed for drug-resistant tuberculosis treatment. They must be carefully selected from a seemingly long list of drugs and regimens. However, patients with multidrug- or extensively drug-resistant tuberculosis prove that this list is still too short.

Drug-resistant Mycobacterium tuberculosis strains complicate treatment protocols, requiring more extensive use of second-line antituberculosis drugs that often involve complex, multi-drug regimens. While necessary for effective treatment, these regimens are linked to a higher incidence of adverse drug reactions [2]. Such reactions can lead to treatment interruption and poorer adherence [3].

Effective management of the side effects associated with antituberculosis therapy requires an understanding of the pharmacodynamics and pharmacokinetics involved, including the mechanisms of drug action and side effects. Clinicians must be able to identify both the common and rare adverse reactions, along with the risk factors that may predispose certain individuals to a heightened risk of these side effects [4].

There have been excellent reviews [2,5,6] published on the topic of adverse drug reactions. Most of them were published some time ago, and new data emerged regarding some of the drugs. There are also new antituberculosis drugs, like bedaquiline, delamanid, and pretomanid, that were unavailable before some of the previously published reviews. We aim to focus on the prevalence of adverse effects in different studies and the risk factors for their development. We also include management and prevention recommendations.

## 2. Adverse Drug Reactions of First-Line Antituberculosis Drugs

Isoniazid, also named isonicotinic acid hydrazide, was synthesized in Prague in 1912 and has been used as an antitubercular drug since 1952. It remains one of the essential components for drug-susceptible and latent tuberculosis treatment to this day. It has recently started to be used in some regimens for drug-resistant tuberculosis [7,8,9,10]. Isoniazid is a nicotinic acid derivative. It passively diffuses into the bacterial cytoplasm and is activated by the enzyme KatG. During activation, isoniazid forms free radical species that disrupt many cellular processes. The isonicotinyl radical binds with the InhA enzyme and NADH. Since the InhA enzyme participates in mycobacteria’s mycolic acid synthesis, this process is blocked, and the cell wall cannot form normally. Radical species also affect nucleic acid, protein, carbohydrate, and lipid synthesis [11,12]. Isoniazid is most effective against actively dividing bacteria [12].

Isoniazid is metabolized in the liver and forms toxic metabolites (e.g., acetyl hydrazine), which are most likely the cause of liver damage [12,13]. Pyridoxine (vitamin B6) metabolism can be disrupted when taking isoniazid, causing reversible neuropathy. While hepatotoxicity is most common within the first few weeks, peripheral neuropathy occurs gradually after several weeks or months [14,15,16,17,18]. The severity of those adverse drug reactions can vary, the most common being elevated liver enzymes and gastrointestinal disturbances [5,15], and the relatively common and more severe—peripheral neuropathy [15]. Severe but less common reactions are symptomatic hepatotoxicity and skin reactions like Stevens–Johnson syndrome and drug-induced lupus [5]. Psychosis, optic neuritis, and pancreatitis are rare, severe adverse reactions [5,19,20]. Through several CYP enzymes, isoniazid can potentially increase concentrations of phenytoin, carbamazepine, warfarin, theophylline, and benzodiazepines [5]. The risk factors for adverse drug reactions due to isoniazid include older age, daily alcohol consumption, and pre-existing liver disease, particularly hepatitis C [21]. However, findings on the relationship between age and hepatitis vary; some studies indicate that neither age nor hepatitis is linked to an increased risk of liver damage [22,23]. It is commonly agreed that the genetic slow acetylator phenotype is a significant risk factor for isoniazid-induced adverse drug reactions to develop [7,15,23,24]. There is a more significant risk for drug-induced peripheral neuropathy in the presence of diabetes, HIV infection, uremia, malnutrition, and alcoholism [14].

Since hepatotoxicity is one of the most common adverse drug reactions, liver enzyme monitoring every 2–4 weeks during the intensive phase should be conducted for those with higher risk and/or baseline abnormalities. To prevent peripheral neuropathy, pyridoxine 50–100 mg daily supplementation needs to be administered. In the case of skin reactions, the first choice of treatment could be ointments with antihistamines and corticosteroids; in severe cases, desensitization may be attempted [4,5,14,24]. When isoniazid-induced mania manifests, the drug should be discontinued; in the more serious cases, psychiatric medications are needed [20].

Imam et al. conducted an observational study involving patients with active tuberculosis who were being treated with first-line antituberculosis drugs. The study revealed a 0.99% discontinuation prevalence of isoniazid as a result of hepatotoxicity, 0.89%—fever; 0.79%—peripheral neuropathy; 0.69%—skin reactions; 0.49%—hypersensitivity; and 0.18%—psychiatric changes [18].

In the studies where isoniazid for latent tuberculosis treatment was used, the prevalence of total adverse drug reactions to isoniazid ranged from 9.2% to 12.8% [25,26,27,28,29,30]. Two studies with the widest participant samples were conducted in the USA and Italy, although many patients were migrants and refugees from South America, Africa, and Asia. Liver enzyme elevation was seen in the USA study for 1.0% and in the Italy study for 5.5% of cases, with hepatotoxicity and severe hepatitis in 1.8% and 0.4%, respectively [29,30].

When used to treat latent tuberculosis, isoniazid caused gastrointestinal disturbances in 1.2–2.8%, skin reactions in 0.4–2.1%, neurological disorders in 1.4–2.5%, peripheral neuropathy in 0.1–0.8%, and psychiatric disorders in 0.7% of cases [15,25,26,27,28,29,30].

Rifampicin (rifampin) was synthesized in 1965 and started to be used in 1968. It belongs to the antibiotic class rifamycin [31]. This antibiotic enters the cytoplasm of bacteria and binds with RNA polymerase, inhibiting RNA synthesis [32]. Rifampicin has high sterilizing activity in bacteria recovering from dormancy and in the case of a spurt of metabolism (actively dividing and semi-dormant bacteria) [33]. Despite the drug’s short half-life, the antibiotic effect remains active through its metabolites [34].

Liver injury caused by rifampicin is uncommon but can occur due to hypersensitivity reactions. In combination with isoniazid, hepatotoxicity is more frequent, since rifampicin activates the CYP3A4 enzyme and thus accelerates the metabolism of isoniazid, resulting in higher rates of hydrazine [13].

The most common reactions are rash, gastrointestinal disturbances, and elevated liver enzymes [5,15]. Less common but more severe adverse drug reactions—hepatotoxicity; thrombocytopenia; acute renal failure; flu-like syndrome; rare hemolytic anemia; and pseudomembranous colitis [2,5]. Gastrointestinal disturbances manifest mainly during the first month of the treatment [13].

Since rifampicin strongly induces cytochrome P450 enzymes and UDP-glucuronosyltransferases and drug transporters, it can lead to decreased plasma concentrations of other drugs [5,34,35]. Caution is needed with antiretrovirals, anticoagulants, anticonvulsants, some antifungals, immunosuppressants, digoxin, beta-blockers, calcium channel blockers, oral hypoglycemics, theophylline, methadone, and others [2,5,36].

Risk factors for rifampicin-induced adverse reactions are not very clear since the drug is seldom used alone [21]. Yee et al. found that age over 60 years and HIV infection were associated with higher rates of adverse drug reactions [37]. Mild gastrointestinal disturbances can be managed symptomatically, but monitoring of liver enzymes is needed to detect hepatotoxicity in time [5]. In the case of thrombocytopenia, the drug should be discontinued; corticosteroids and platelet transfusions, in some cases, are needed [38]. Mild hypersensitivity reactions might need antihistamines or corticosteroids; in severe cases, desensitization may be attempted [39]. The frequency of treatment discontinuation due to adverse drug reactions varies among studies, in 0.6–1.19% [15,18].

In a study of latent tuberculosis treatment, rifampicin showed a lower prevalence of adverse drug reactions as compared to isoniazid. Elevated liver enzymes were seen in 0.8%, hepatotoxicity in 0.08%, gastrointestinal disturbances in 2.4%, and rash and skin reactions in 1.6% of cases. A rare but significant adverse drug reaction was thrombocytopenia, which was seen in 0.2% of cases [29]. Another latent tuberculosis treatment study showed a much higher prevalence of adverse drug reactions: gastrointestinal disturbances in 22.7%, skin reactions in 5.1%, and neurological symptoms in 9.6% of cases [26].

When used in combination, data on adverse risk factors attributed to rifampicin vary. Castro et al. report only rash and gastrointestinal disturbances due to rifampicin, both as frequent as 1.6% [40]. Another study found that rifampicin was a culprit of hepatotoxicity in 0.2%, cutaneous reactions in 0.1%, flu-like syndrome in 0.26%, hemolytic reaction in 0.1%, and change in glucose tolerance in 0.1% of cases [15]. Yee et al. found that rifampicin was the cause of a rash in 2% and gastrointestinal disturbances in 1% of cases [37].

Pyrazinamide, as a chemical compound, has been known since 1931. Still, its role in tuberculosis treatment was only recognized in 1952 when other nicotinamide derivatives were recognized as effective drugs against *M. tuberculosis* [41]. Pyrazinamide is a prodrug that is converted into pyrazinoic acid by the intracellular enzyme of *M. tuberculosis*, which damages cellular walls by inhibiting fatty acid synthesis [41,42,43,44,45]. Its activity against dormant, nongrowing mycobacteria in tuberculosis treatment is irreplaceable, though perplexing—the mechanism of action is still poorly understood [41].

It is widely accepted that pyrazinamide is hepatotoxic. At the same time, the mechanisms of its toxicity are not precisely known, and there is a lack of studies that could collect prevalence data on pyrazinamide toxicity since it is mainly used together with other drugs. However, data shows that pyrazinamide hepatotoxicity is a direct toxic effect through its metabolites, not related to hypersensitivity or immune response [46,47]. Results on dose dependency differ throughout studies [46,48]. Pyrazinamide can inhibit urine acid transporters in the kidneys, resulting in the reuptake of uric acid back into the bloodstream. This causes hyperuricemia, joint pain, and kidney stones if oxalates are formed [49,50].

Hepatotoxicity can occur at any time during treatment, but it is most common during the first 2–9 weeks, as is hyperuricemia [15,24,50]. Adverse reactions due to pyrazinamide can vary from mild to severe. More common are hyperuricemia, arthralgia, elevated liver enzymes, exanthema, and gastrointestinal disorders; less common but more severe are hepatotoxicity, gout, and rhabdomyolysis [5,24,50]. Rarely, toxic epidermal necrolysis can develop [51].

Not much is known about drug interactions—pyrazinamide can reduce the effectiveness of allopurinol by increasing uric acid levels [52]. It antagonizes the effects of probenecid and decreases the serum concentration of cyclosporine [5].

Older age and diabetes mellitus were found as risk factors for pyrazinamide-induced hepatotoxicity in one study [48], but in others, it was not confirmed [15]. As the half-life of pyrazinamide is significantly prolonged in patients with pre-existing liver or kidney disease [46], one could account for those as risk factors for adverse drug reactions to develop. Risk factors for uricemia include gout or elevated uric acid levels before antituberculosis treatment [53].

In the case of hyperuricemia and related symptoms, allopurinol should be administered [5,52]. Arthralgia can be managed with analgesics [21,24]. If significant liver injury is present, pyrazinamide should be discontinued [5,21,24].

Pyrazinamide discontinuation rates vary. Some studies report rates of 2–5% [37]. However, in elderly patients, early discontinuation of pyrazinamide is frequent, 20.6% [48]. In a study from Turkey, pyrazinamide had to be stopped due to severe hepatotoxicity in 7 cases (0.6%) [15].

Borisov et al. reported adverse drug reactions due to pyrazinamide in 13.2% of cases, most of which were mild (11.3%) [54]. In a South Korean study, the prevalence of hepatotoxicity was 5.13%, and 10 out of 12 drug-induced hepatotoxicity cases were pyrazinamide-related [55]. In the study of Shin et al., which had a significant sample of participants with a wide range of concomitant diseases, 87% of participants treated with pyrazinamide had hyperuricemia [56]. Smaller sample studies showed a prevalence of 58–60% [50,53], and one study from India showed 28.57% [57].

Ethambutol was mentioned for the first time in 1961 by the Lederle Company (New York, NY, USA) when they announced the discovery of a new synthetic compound that protects mice from the lethal *M. tuberculosis* strain [36]. Ethambutol targets Emb proteins, competing for binding to the EmbB and EmbC subunits. These enzymes are important in cell wall biosynthesis [58].

Optic neuropathy is the primary concern when prescribing ethambutol. The precise mechanism is unclear, possibly involving mitochondrial toxicity and disruption of axonal transport in the optic nerve. Optic neuropathy develops after several weeks or months and is rarely seen earlier in treatment [59]. Other less common adverse drug reactions include peripheral neuropathy, rash, and gastrointestinal disturbances. It is known that aluminum-containing antacids may decrease ethambutol absorption [60]. Ethionamide can exacerbate the toxic effects of ethambutol [5].

Risk factors for developing optic neuropathy are high doses of ethambutol, long treatment duration, renal impairment (causing higher levels of ethambutol in the blood), older age, and pre-existing optic nerve disease [5,36,59]. Although optic neuropathy due to ethambutol is reversible in most cases, the drug should be immediately discontinued if signs of this adverse drug reaction occur [21,24]. Visual acuity and fields should be monitored, and ophthalmologist consultation is needed [4,59]. Discontinuation rates due to ethambutol-related adverse reactions are variable, 0.2–1% when 15 mg/kg doses are prescribed [5,21] and 15–18% at a 35 mg/kg dosage [5,36].

The most common drug combination for drug-susceptible tuberculosis treatment includes two months of rifampicin, isoniazid, pyrazinamide, and ethambutol for the intensive phase, followed by four months of rifampicin and isoniazid alone [9]. The prevalence of adverse drug reactions with combined drug regimens varies significantly between studies, ranging between 8 and 83 percent of patients who were treated with first-line antituberculosis drugs using the Directly Observed Treatment and Short-course chemotherapy strategy [2,61,62,63]. There is an even more significant difference if you take studies that did not use those strategies into account [64]. The absolute majority of adverse drug reactions were reported in the intensive treatment phase [65,66,67]. Most of them, during the first month of treatment [68], and some even in the first week [67].

A recent retrospective study used data from a spontaneous adverse event reporting system in Korea; there were almost 18 thousand cases with a prevalence of adverse drug reactions to first-line antituberculosis drugs of 1.14%. All four drugs were similarly common causes of those reactions, with 28.7% for rifampicin, 24.0% for isoniazid, 23.4% for ethambutol, and 23.9% for pyrazinamide. Approximately 80.37% of adverse drug reactions were possible, 17.54% probable, and 2.09% certain. Nausea and liver enzyme elevation were most common, with a prevalence of 14.6% and 14.2%, respectively. Rash (11.7%), pruritus (9.1%), and vomiting (8.9%) were also among the most commonly reported reactions [68].

However, a prospective cohort study in Brazil showed a 78.8% prevalence of adverse drug reactions to first-line antituberculosis drugs [65]. In another prospective observational study from India, a prevalence of 34.72% of adverse drug reactions was found; 10.09% of them were definite, 5.83% probable, 12.17% possible, and 6.63% doubtful [18].

Significant variation in adverse drug reaction prevalence shows the importance of study design, sample size, terminology, and characteristics of participants.

Moxifloxacin- and rifapentine-based treatment combinations are now available for drug-susceptible tuberculosis treatment [9]. In a multicenter randomized controlled trial, the significant difference between the rifapentine–moxifloxacin group and the control group according to grade 3 and higher adverse drug reaction prevalence was not found; it was 18.8% and 19.3%, respectively [69].

Mechanisms of drug action and adverse drug reactions, drug interactions, and risk factors for adverse drug reactions to develop are summarized in Table 1.

Timing, frequency, monitoring and management, and discontinuation rates due to adverse drug reactions are summarized in Table 2.

## 3. Adverse Drug Reactions of Second-Line Antituberculosis Drugs

Fluoroquinolones are fluorine-containing nalidixic acid derivatives, first synthesized in 1962 [71]. It has broad-spectrum antimicrobial activity by inhibiting DNA gyrase, which is a supercoiling enzyme. This enzyme is essential for DNA expression and replication. When DNA ends are free, mRNAs, exonuclease, and some proteins are produced uncontrollably, and chromosomes start to degrade. This leads to the death of bacteria [6]. In the latest World Health Organization recommendations, levofloxacin (third-generation fluoroquinolone) and moxifloxacin (fourth-generation fluoroquinolone) are in group A of drugs for drug-resistant tuberculosis treatment [10].

Fluoroquinolones can block cardiac potassium channels, leading to prolongation of the QT interval [72]. Due to those drugs, tendon rupture can occur, particularly in the Achilles tendon, but the mechanism is not fully understood. It may involve the production of reactive oxygen species and matrix metalloproteinase activation, weakening tendons [73].

In one study, adverse drug reactions possibly to moxifloxacin occurred in a median of 15 days and to levofloxacin in a median of 35 days [74]. Tendon rupture can occur early during treatment and even weeks to months after treatment [73]. More common adverse drug reactions are gastrointestinal disturbances, nausea, diarrhea, dizziness, insomnia, and skin rash, while less common but more severe reactions are QT prolongation, arrhythmia, tendon rupture, peripheral neuropathy, hallucinations, delusions, pseudomembranous colitis, urticaria, and vasculitis [2,6,73,75].

Possible fluoroquinolone drug interactions include other QT-prolonging drugs (e.g., amiodarone, erythromycin) [76] and drugs and supplements containing multivalent cations—aluminum; magnesium; calcium; and iron (laxatives, some antacids, etc.). Concomitant use of fluoroquinolones can affect oral anticoagulant blood levels. Probenecid and cimetidine can increase levels of fluoroquinolones. When used with analgesics, the possibility of convulsions is higher [6]. Risk factors for QT prolongation are elderly, female sex, electrolyte imbalance, cardiac diseases, and use of other agents that prolong the QTc interval [2,77]. Tendon rupture is more common in patients who are older, have renal insufficiency, and are taking corticosteroid therapy [78].

ECG monitoring is needed to detect adverse drug reactions of fluoroquinolones early, especially in high-risk patients. It is crucial to correct electrolyte imbalances if there are any. If significant QT prolongation or arrhythmia occurs, treatment should be discontinued [2,76].

In a meta-analysis of individual-level patient data of over nine thousand patients, adverse drug reactions that resulted in permanent antituberculosis treatment discontinuation were analyzed. The pooled absolute risk for levofloxacin (3.2%) and moxifloxacin (4.5%) was low [79,80].

In a retrospective study from South Korea, levofloxacin was accountable for slightly more adverse drug reactions than moxifloxacin (11.0% vs. 8.2%). In the levofloxacin group, gastrointestinal disturbances were the most common problem, affecting 5 of 82 participants, followed by musculoskeletal, neurological, and renal adverse reactions; hepatotoxicity; and allergic reactions—each affected one participant. Meanwhile, in the moxifloxacin group, allergic reaction was the main problem, affecting 5 out of 122 participants; gastrointestinal disturbances and neurological problems were seen in 2 participants; in one case, hepatotoxicity was reported [74]. Fluoroquinolones are known to increase the QT interval; moxifloxacin has a more significant effect on ECG changes than levofloxacin [76,81]. Another worldwide study reported opposite results—adverse drug reactions occurred in 6.8% and 10.3% due to levofloxacin and moxifloxacin; respectively [54].

Linezolid belongs to the antimicrobial group called oxazolidinones; they were first used in 1978 for plant diseases. The US Food and Drug Administration approved linezolid in 2000. This drug disrupts protein synthesis by binding to rRNA and inhibits the initiation process for protein synthesis, disrupting protein elongation [82]. Linezolid is in a multidrug-resistant tuberculosis treatment drug group A, according to World Health Organization guidelines [10].

In bone marrow cells, linezolid can disrupt mitochondrial function. Mitochondrial toxicity is the probable cause of peripheral and optic neuropathy, though the mechanism of those adverse drug reactions is not fully understood. The mechanism of myelosuppression is not known. It was thought that it had the same mitochondrial toxicity, but some studies show normal bone marrow, indicating a direct toxic effect on blood cells.

Myelosuppression typically occurs after 2–4 weeks of initial treatment. Peripheral neuropathy develops gradually after several weeks or months. Optic neuropathy can develop after more than 2 months of treatment [82,83]. The most common adverse reactions are gastrointestinal disorders, peripheral neuropathy, and anemia; less common are optic neuritis, thrombocytopenia, and lactic acidosis [2,24]. Optic neuritis due to linezolid is irreversible in most cases [2].

Linezolid has no interactions with most antimicrobials, but there is not enough evidence about interactions with rifampicin. When combined with serotonin reuptake inhibitors, linezolid can cause life-threatening toxicity. It is not contraindicated to prescribe those drugs together, though [82]. Risk factors for thrombocytopenia are baseline platelet count, minimum concentration, and renal insufficiency [84].

Plasma concentrations of linezolid are not dependent on age or mild to moderate hepatic or renal failure [82]. No other risk factors are known for adverse drug reactions. If peripheral neuropathy or optic neuritis occurs, treatment should be discontinued. Meanwhile, anemia and thrombocytopenia could be reversed with a dosage reduction [24].

Individual-level patient data meta-analysis showed that linezolid caused adverse drug reactions, which led to the discontinuation of the drug in 14.1% [79].

Another meta-analysis showed 58.9% of adverse drug reactions were due to linezolid, and 68.4% of them resulted in treatment interruption or dose reduction. The most common were anemia (38.1%) and peripheral neuropathy (47.1%); other adverse reactions were gastrointestinal disorders in 16.7%, optic neuritis in 13.2%, and thrombocytopenia in 11.8% of cases. More adverse drug reactions were seen in daily linezolid dosages >600 mg as compared to ≤600 mg (74.5% vs. 46.7%) [85]. However, another study reported linezolid-induced adverse reactions in 16.2% of patients, most of them (14.6%) were mild [54].

Clofazimine, a hydrophobic riminophenazine, was synthesized in 1954 as an antituberculosis drug. At first, it was thought to be ineffective. However, the drug showed good anti-leprosy results [86,87,88]. The first theory of the mechanism of action of clofazimine was that the drug binds to the guanine amino acid of bacterial DNA, which results in inhibition of bacterial proliferation [89]. Still, new findings show that the drug increases reactive oxidant species and destabilizes the bacterial membrane. Clofazimine reverses the inhibition of intracellular phagocyte killing mechanisms and acts synergistically with interferon-gamma [90,91,92,93].

Skin discoloration develops due to its bioaccumulation and partitioning into subcutaneous fat [94]. Previously, this color was thought to come from crystal-like structures inside macrophages. Clofazimine strongly inhibits hERG cardiac potassium channels, which results in QT prolongation and a higher risk for arrhythmias and sudden cardiac death [95].

Scientific articles cannot be found that report the time of onset of adverse drug reactions. The clofazimine’s FDA label says skin discoloration develops within a few weeks. No clinically significant differences in clofazimine pharmacokinetics have been observed when used concomitantly with bedaquiline, cycloserine/terizidone, dapsone, ethionamide, para-aminosalicylic acid, pyrazinamide, and pyridoxine. 

Borisov et al., in their worldwide study, found 7.0% of total adverse reactions due to clofazimine, including 1.4% serious and 5.6% mild [54].

The most common adverse drug reaction caused by clofazimine is yellow to brownish skin discoloration, which occurs in 75–100% of cases [2,24,86,96]. It is reversible, though the drug has a long-lasting effect. In 40–50% of patients, gastrointestinal disturbances occurred. Skin dryness and ichthyosis can also develop in 8–28% [2,86]. QTcF prolongation is another profound side effect of clofazimine, especially important when tuberculosis treatment includes fluoroquinolones, bedaquiline, or delamanid [97]. Less common possible adverse reactions—hepatitis; hypersensitivity reaction; nephrotoxicity; and acne [2].

If skin discoloration occurs, the patient should be advised that this adverse drug reaction will be resolved over a few months or years after discontinuing the drug. Gastrointestinal disturbances can be managed symptomatically in most cases [24]. When QTC prolongation develops, analysis for cardiac diseases and electrolyte imbalance should be performed [2].

In a systematic review by Gopal et al., the prevalence of adverse drug reactions attributable to clofazimine was 11.4%. The drug was discontinued in <1% of patients [86]. Most of the studies in this review were conducted in Asian and South American countries. In a worldwide meta-analysis, the occurrence of discontinuation of clofazimine due to adverse reactions was 1.6% [79].

Cycloserine is one of the oldest antituberculosis drugs, developed around 1955 [88]. It inhibits D-alanyl-D-alanine synthetase, alanine racemase, and alanine permease. These are the enzymes crucial in peptidoglycan formation for the bacterial cell membrane [6,98]. Terizidone is a structural analog of cycloserine; it is a prodrug—each terizidone molecule contains two cycloserine molecules [99].

Cycloserine/terizidone acts as a partial NMDA receptor agonist, which may contribute to its neuropsychiatric effects. The exact mechanism of many CNS and psychiatric cycloserine/terizidone-induced adverse drug reactions is not fully understood [100]. One study found that adverse drug reactions appeared after a median of 71 days [101].

The most common adverse drug reactions due to cycloserine/terizidone are headache and tremors, sleep disturbances, anxiety, depression, confusion, and pale skin. Less common symptoms—visual changes; skin rash; hepatitis; tingling; and numbness in the extremities [2]. In a meta-analysis by Hwang et al., the cycloserine/terizidone pooled estimated prevalence for adverse drug reactions was 9.1%:5.7% psychiatric and 1.1% central nervous system [102]. Borisov et al. reported total adverse events in 6.0% of cases (1.8% serious and 4.4% mild) [54]. Another retrospective study from China showed a 4.3% prevalence of psychiatric adverse drug reactions attributable to cycloserine/terizidone [103].

Risk factors for adverse reactions to develop are higher doses and longer duration [24,104]. When combined with ethionamide and isoniazid, neurotoxic effects can be more frequent. Cycloserine/terizidone can increase serum levels of phenytoin and oral anticoagulants [6].

If seizures or psychotic symptoms develop, the drug should be discontinued. Pyridoxine supplementation in moderate doses is recommended to lower the risk of neuropathy and CNS toxicity [2].

A meta-analysis of individual patient data showed that cycloserine/terizidone was permanently discontinued because of psychiatric adverse drug reactions in 66% of patients [79]. Another study found a discontinuation prevalence of 6.25% [101].

Carbapenems—a class of β-lactam antibiotics; including meropenem and imipenem; that are recommended in some cases of drug-resistant tuberculosis treatment [10]. Thienamycin—the first carbapenem and model for all carbapenems—was discovered in 1976 [105]. There is minimal evidence of carbapenem efficacy, safety, and tolerability in tuberculosis treatment. This group of antibiotics enters the bacterial cell and acylates the PBPs enzymes, which catalyze peptidoglycan formation. This causes continued autolysis and cell damage due to osmotic pressure [105]. *M. tuberculosis* produces the enzyme β-lactamase, which can break down some β-lactam antibiotics, including meropenem. Still, while slowly broken down, it acts as a β-lactamase inhibitor, allowing other antibiotics to work. If combined with clavulanic acid to block β-lactamase, meropenem is more effective [106].

Carbapenems can bind to GABA receptors, causing seizures [107]. As with most broad-spectrum antibiotics, carbapenems disrupt gut flora, elevating the risk of developing pseudomembranous colitis.

There is no data on the timing of adverse drug reactions of carbapenems related to tuberculosis treatment. Considering the mechanism of action and adverse drug reactions, gastrointestinal symptoms and injection site reactions could occur early, while seizures and pseudomembranous colitis should take more time to develop.

The most common adverse drug reactions are gastrointestinal disturbances like nausea, vomiting, and abdominal pain, which occur in up to 20% of patients. Less common—pseudomembranous colitis. Rare reaction—seizures; reported in 1.5% of patients; mostly when given high doses; is more linked to imipenem [70,107]. When administered for a long time, injection site inflammation can occur (in 1.1% of patients) [24,107].

Overall, carbapenems have a low potential for drug interactions. When used with ganciclovir, the risk for convulsions elevates. Imipenem lowers the serum concentration of valproate [70].

There is a higher risk of seizures for those who have a history of them [107]. No specific data on the risk factors of carbapenem-induced adverse drug reactions were found.

Management of gastrointestinal symptoms is symptomatic. When diarrhea and fever occur, patients should be tested for pseudomembranous colitis [70].

Sotgiu et al. found up to 15% prevalence of adverse drug reactions in their systematic review [108]. Meanwhile, the pooled incidence of adverse drug reactions was 7.7% in drug-resistant tuberculosis treatment in a meta-analysis conducted by Lan et al. Meropenem and imipenem were permanently discontinued in 4.9% of cases due to adverse drug reactions [79].

Amikacin is a semi-synthetic derivative of kanamycin A, approved in 1976. Aminoglycosides penetrate into the bacterial cytoplasm and interfere with the translation of proteins, damaging the cytoplasmic membrane [109].

Permanent bilateral hearing loss due to damage to cranial nerve VIII is a severe adverse drug reaction of aminoglycosides; the risk of it increases with age, duration of treatment, and accumulated dose [6,110]. Nephrotoxicity is also a concern because of the accumulation of the drug in renal tubules [6,111].

Most amikacin-induced adverse reactions are duration-dependent when the accumulated dose is high. More frequent adverse events are pain at the injection site and proteinuria; more severe but less common are cochlear, vestibular, and nephrotoxicity; peripheral neuropathy; rash; and eosinophilia [2].

Borisov et al. reported adverse drug reactions due to amikacin in 22.9% of cases in the overall cohort, including 6.9% grades 3–5 and 16.0% grades 1–2 [54]. In the Lan et al. meta-analysis, a pooled incidence of amikacin-related adverse drug reactions was 13.8%. Ototoxicity was the most common among adverse drug reactions of amikacin, causing drug discontinuation in 87% of those cases [79].

Ototoxicity and nephrotoxicity can be more frequent when amikacin is used with amphotericin B, vancomycin, cephalosporin, cisplatin, and loop diuretics. Concomitant use of neuromuscular blocking agents can cause respiratory depression due to muscle weakness [6].

Risk factors for ototoxicity are age, long treatment duration, high total accumulated dose, concomitant usage of diuretics, dehydration, and history of hearing impairment. Monitoring with audiometry during treatment is needed. If hearing loss is seen, a decision weighing the risks and benefits of the treatment should be made—in some cases; the drug must be discontinued; as it should be if nephrotoxicity develops. Electrolyte imbalance must be corrected [2].

Nephrotoxicity is more common in older patients with a history of kidney disease. Neuromuscular blockades often develop in patients with hypocalcemia, hypokalemia, hypomagnesemia, and in the presence of botulism or myasthenia gravis [6].

Ethionamide was started to be used in medical practice around 1960 [112]. Together with prothionamide, it belongs to the thioamide group of drugs; its structure is similar to that of isoniazid. Thioamides are prodrugs that are activated inside bacterial cells. Like isoniazid, the primary target of ethionamide is InhA, resulting in inhibition of mycolic acid biosynthesis. However, activation of these drugs differs—it was suggested that ethionamide is activated by an enzyme, EthA [80,113,114].

Since ethionamide is similar to isoniazid, adverse drug reactions are similar, too. Hepatic adverse reactions can occur for up to five months after the start of the treatment.

Gastrointestinal disturbances are common and severe when prescribing ethionamide. It includes metallic taste, excessive salivation, nausea, vomiting, loss of appetite, and abdominal pain. Less common are hepatotoxicity, neurological and psychiatric disturbances, hypothyroidism, menstrual irregularity, gynecomastia, arthralgia, and leukopenia. Rare adverse reactions—peripheral and optic neuritis; rash; photosensitivity; and thrombocytopenia [2,6].

When ethionamide is used with terizidone or isoniazid, neurotoxic reactions are more common. When used with para-aminosalicylic acid, hepatotoxicity and risk for hypothyroidism are higher. When used with alcohol, psychotic reactions can develop [6].

Risk factors for hepatotoxicity are a history of liver disease and alcoholism. In the history of mental instability, ethionamide should be administered with caution. Pyridoxine supplementation is recommended when prescribing ethionamide [6].

Symptomatic treatment is needed for gastrointestinal reactions; sometimes, doses of ethionamide can be increased progressively. When psychiatric symptoms occur, specialist consultation may be required. In some cases, drugs can be stopped, and psychiatric medications prescribed [2].

A study showed adverse drug reactions in 17.6% of patients (0.4% serious and 17.2% mild) [54]. In a meta-analysis conducted by Lan et al., the summed pooled incidence of adverse drug reactions was 10.9%. When ethionamide and prothionamide were discontinued, in most cases, it was due to gastrointestinal disorders (48%) [79].

Para-aminosalicylic acid (PAS) has been one of the first antituberculosis drugs since 1946. Despite a long time in clinical use, the mechanism of action remained elusive for a long time. PAS acts as a competitive inhibitor of para-aminobenzoic acid in the folate synthesis by targeting dihydropteroate synthase. This leads to inhibition of dihydrofolate reductase and impaired bacterial growth. Additionally, PAS metabolites can inhibit thymidylate synthase, further disrupting DNA synthesis [115,116].

The mechanism of PAS-induced adverse drug reactions is not fully understood; they can develop due to immune responses to the drug or its metabolites, thyroid hormone synthesis interference, and drug impact on intestinal function (reduces vitamin B12 absorption) [70,117,118].

Gastrointestinal intolerance occurs after one week of treatment or more [119].

The prevalence of adverse drug reactions attributed to PAS varies from 10 to 30% [118]. Common reactions are gastrointestinal disturbances and hypothyroidism. The prevalence of gastrointestinal disturbances when using PAS was reported in 12–58% of cases and was dose-dependent. Hypothyroidism may develop in 40% of the patients. Less common are reactions—hepatitis (0.3–0.5%); allergic reactions (fever, rash, pruritus) (5–10%); hemolytic anemia; granulocytopenia; polyneuritis; pericarditis; malabsorption; etc. [6,24,118,120]. Most of those findings are from studies conducted many years ago, which reflects the need for newer data.

When PAS is prescribed with rifampicin, the level of rifampicin in the blood can fall by about half [121]. Caution is needed when using PAS with probenecid, sulfonylurea, oral anticoagulants, thrombolytics, and salicylates [116].

There is not enough data on risk factors for adverse drug reactions, precisely due to PAS development. Concomitant use of PAS with ethionamide, prothionamide, and some nonselective NSAIDs can lead to more frequent gastrointestinal disturbances. These reactions are also more frequent when higher doses of PAS are prescribed—12%; 15%; and 52% of patients experienced some symptoms when given 5 g; 10 g; or 20 g of PAS per day; respectively [118,120,122].

Occasionally, diarrhea improves after several weeks of treatment, and nausea and vomiting can be managed symptomatically. Thyroid function normalizes after drug discontinuation [24]. Until then, thyroxine therapy should be initiated. Sometimes, splitting the dose or timing with food alleviates symptoms [2].

Severe adverse drug reactions occurred in 0.5% of cases in the worldwide study, with mild reactions in 10.7% [54]. Treatment interruption is needed in 4% [24], and permanent discontinuation—in 11.6% of cases [70].

Bedaquiline is a diarylquinoline group antituberculosis drug discovered in 2005 [123]. It inhibits mycobacteria-specific F-ATP synthase by binding to the c subunit. This halts ATP production and leads to bacterial death [124]. Bedaquiline is active against dormant and actively replicating mycobacteria [125].

Bedaquiline inhibits cardiac hERG potassium ion channels, and this can lead to QTc prolongation [126]. Hepatotoxicity involves mitochondrial dysfunction and alterations in cellular signaling pathways [127].

Concerning timing, the most significant increase in QTc was seen in the first 6 weeks after bedaquiline initiation in a retrospective study by Isralls et al. [128]. Other data shows that QTc increased by 10–15 msec, reaching a maximum at week 15 [70]. Wu et al. analyzed the US Food and Drug Administration’s Adverse Event Reporting System’s reports about bedaquiline. They found that 34.2% of reactions were seen in the first month and 14.2% in the second month [127].

More common adverse drug reactions include nausea, vomiting, abdominal pain, anorexia, arthralgia, headache, and QTc prolongation. Rarely do hyperuricemia, phospholipidosis, and elevated transaminases occur. Elevated liver enzymes are a sign of an increased risk of pancreatitis [2,70].

When bedaquiline is used with rifamycin, it may cause a significant reduction in bedaquiline concentration. Azole antifungals and macrolides can increase concentration. Caution is also needed when taking bedaquiline together with efavirenz, phenytoin, glucocorticoids, metoclopramide, furosemide, hydrochlorothiazide, citalopram, escitalopram, methadone, antiarrhythmics, and also fluoroquinolones, clofazimine, and delamanid—drugs that are also associated with QTc prolongation [70].

Some authors suggest discontinuing treatment with bedaquiline if liver enzymes are >5 × ULN or >3 × ULN with symptoms. Less likely to cause drugs should be reintroduced when <2 × ULN, adding one drug at a time every 3 days [2,24]. Gastrointestinal symptoms and arthralgia can be managed symptomatically and improve after a few weeks of treatment. Serial monitoring with ECG is recommended [2].

Lan et al. found that bedaquiline has one of the lowest incidences of adverse drug reactions that led to discontinuation of the drug (1.7%) [79].

Bedaquiline-associated reactions developed in 11.1% of cases in the Borisov et al. study (serious—1.0%, mild—10.1%) [54]. Adverse drug reactions that occurred in bedaquiline-containing regimens are hepatotoxicity (pooled rate, 12.6%), renal disorders (pooled rate, 5.9%), optic neuropathy, including blurred vision (pooled rate, 3.9%), ototoxicity, including hearing loss (pooled rate, 7.0%), hematological disorders (pooled rate, 12.5%), gastrointestinal symptoms like nausea or vomiting (pooled rate, 13.8%), peripheral neuropathy (pooled rate, 13.9%), electrolyte disturbances (pooled rate, 6.4%), arthralgia (pooled rate, 10.1%), psychiatric disorders (pooled rate, 4.6%), and dermatological disorders including acne (pooled rate, 9.8%) in Rehman et al.’s meta-analysis. The QTc interval was prolonged in 10.2% of cases when treatment included bedaquiline [129].

In an observational cohort from Guglielmetti et al., gastrointestinal side effects were most common (71.7%), followed by oto-vestibular impairment (55.6%) and peripheral neuropathy (40.9%). No differences were found between standard and prolonged bedaquiline treatment groups [130].

Delamanid was approved in 2014 for the treatment of multidrug-resistant tuberculosis [131]. It is a prodrug that belongs to the dihydro-nitroimidazole class. Delamanid disrupts the mycobacterial cell wall by inhibiting the synthesis of methoxy-mycolic and keto-mycolic acids. During activation by the enzyme deazaflavin-dependent nitroreductase, delamanid produces reactive nitrogen species, which further disrupt bacterial metabolic processes [132,133,134].

The metabolism of delamanid is not fully understood; the drug is probably converted to the primary metabolite DM-6705. This metabolite is associated with delamanid toxicity, mainly with QTc prolongation [135].

Changes in ECGs peak at week 8; average QTc prolongation is 5–15 ms.

Delamanid is well tolerated; the primary concern when using this drug is QTc prolongation. Other possible reactions are nausea, vomiting, dizziness, insomnia, anxiety, hallucinations, night terrors, and upper abdominal pain [2,24,70].

Caution is recommended when using delamanid together with rifamycin, carbamazepine, ritonavir, ketoconazole, and drugs that tend to prolong QTc themselves—fluoroquinolones, clofazimine, bedaquiline, macrolides, metoclopramide, efavirenz, furosemide, hydrochlorothiazide, citalopram, escitalopram, methadone, antiarrhythmics, etc. When administered with cycloserine/terizidone, there is a higher risk of neuropsychiatric adverse events [70].

At first, it was thought that hypoalbuminemia is linked with an increased risk for QTc prolongation, but recent studies suggest no association [24,70].

In mild cases, a symptomatic approach is recommended for gastrointestinal symptoms. ECG monitoring is recommended [2].

Gler et al. reported similar discontinuation rates in the 100 mg and 200 mg delamanid groups and placebo—2.5%; 3.8%; and 2.5%; respectively. In the 100 mg group, 2 cases were discontinued due to psychiatric side effects, one due to dermatologic side effects, and one due to thrombocytopenia. In the 200 mg group, 1 case was discontinued due to leucopenia, 3 due to psychiatric side effects, and 1 due to dermatologic side effects [132].

Data from the same study showed that in the 200 mg twice daily group, QT interval prolongation occurred more often (13.1%) than at 100 mg twice daily (9.9%). In a placebo group, the prevalence of QT prolongation was 3.8% [132].

Other adverse drug reactions that were common in delamanid treatment groups (lower and higher doses, accordingly) were gastrointestinal disturbances (nausea 36.0–40.6%, vomiting 29.8–36.2%, upper abdominal pain 22.5–25.5%), nervous system (headache 22.4–25.6%, paresthesia 10.6–12.5%, tremor 10.0–11.8%, insomnia 26.1–31.9%), and other (tinnitus 9.9–13.8%, asthenia 12.4–16.9%, malaise 7.5–10.0%, anorexia 14.3–21.2%, hyperuricemia 19.3–23.8%, hypokalemia 12.4–19.4%). Those adverse reactions were also seen in the placebo group; most had a higher prevalence in the high-dose delamanid group [132]. Severe adverse drug reactions were seen in 0.8% of patients and mild in 12.4% [54].

Pretomanid was approved for tuberculosis treatment in 2019 [136]. It is bactericidal against replicating and non-replicating *M. tuberculosis* [123]. Activation of the prodrug is similar to that of delamanid. Pretomanid is transformed into three metabolites after activation. It has two mechanisms of action, depending on conditions in the human body. In an aerobic setting, it inhibits protein and lipid synthesis, decreasing the availability of mycolic acids. In an anaerobic state, pretomanid generates des-nitro metabolites and releases nitric oxide, resulting in a significant reduction of ATP concentrations in cells [137].

Most evidence available about pretomanid toxicity comes from in vitro and preclinical studies; there is no data yet about long-term safety. Animal studies show hepatic, ophthalmologic, and reproductive organ damage when pretomanid is given at the most relevant dose [138].

Common adverse drug reactions are nausea and vomiting, acne, headache, musculoskeletal pain, and liver enzyme elevation [139,140]. The frequency of adverse drug reactions in clinical trials was this: gastrointestinal disturbances in 28.4%, hepatic disorders in 25.5%, liver enzyme elevation in 19.2%, skin reactions in 16.6%, and headache in 11.0% [141]. Pretomanid alone has minimal liver toxicity (2.2%) [138].

When taken with strong CYP3A4 inducers (rifampicin, efavirenz), pretomanid blood levels can be significantly lowered. Meanwhile, with mild inducers (lopinavir/ritonavir), the effect on drug levels is smaller [138].

There is no data about the timing and risk factors of adverse drug reactions. Management recommendations are not put in the WHO operational handbook or other authors’ reviews [2,24,70]. The treatment discontinuation rate due to pretomanid is not known.

New regimens for drug-resistant tuberculosis treatment are now in official World Health Organization guidelines: the 6-month bedaquiline, pretomanid, linezolid, and moxifloxacin (BPaLM) and the 9-month all-oral regimen [10].

In an open-label, multicenter, randomized, controlled trial, there were fewer grade 3 or more adverse drug reactions in the BPALM group than in the standard care group (19% vs. 59%). In both groups, the most common reactions were hepatic disorders (4% vs. 11%), QTcF prolongation (1% vs. 14%), peripheral neuropathy (9% vs. 19%), decreased creatinine clearance (1% vs. 7%), anemia (3% vs. 8%), and neutropenia (3% vs. 8%). Ten patients (2%) died; seven of them were in the standard care group. Four of the deaths were considered to be treatment-related, all of them in the standard care group [142].

The 9-month all-oral regimen consists of bedaquiline (6 months), in combination with levofloxacin/moxifloxacin, ethionamide, ethambutol, isoniazid (high-dose), pyrazinamide, and clofazimine (for 4 months, with the possibility of extending to 6 months if the patient remains sputum smear positive at the end of 4 months), followed by treatment with levofloxacin/moxifloxacin, clofazimine, ethambutol, and pyrazinamide (for 5 months). Ethionamide can be replaced by 2 months of linezolid (600 mg daily). Data concerning the adverse drug reactions for a regimen with ethionamide were not collected, which means that it cannot be compared with a regimen containing linezolid [10].

In the Nix-TB trial, 57% of participants had adverse drug reactions. More than 80% had peripheral neuropathy, and 37% had anemia. Moderate or severe liver enzyme elevation was seen in twelve (11%) participants; no participants had QT interval prolongation of more than 480 ms. In the 9-month regimen, the most common adverse drug reactions were anemia (linezolid-containing regimen), hepatotoxicity, QT prolongation, nausea, and vomiting [70].

## 4. Prevention of Adverse Drug Reactions

There is limited data on the prevention of adverse reactions to tuberculosis medications. Given that alcoholism can lead to liver damage and poses a significant risk for adverse drug reactions, it is crucial to discontinue alcohol consumption as soon as possible. Additionally, pyridoxine supplementation is recommended when prescribing certain antituberculosis drugs. It is essential to meticulously consider drug interactions when selecting medications for tuberculosis treatment, as some drugs can amplify each other’s adverse effects (refer to Table 2 for risk factors, management recommendations, and potential drug interactions).

## 5. Conclusions

Adverse drug reactions are common in tuberculosis treatment with first- and second-line antituberculosis drugs, with a wider variety and more severe reactions seen in the latter. The severity of adverse drug reactions varies considerably; most of them can be managed, and no permanent discontinuation is needed. There is a lack of data on the newest and some old drugs’ adverse reaction mechanisms; most of the evidence about risk factors for drug side effects is from a very long time ago. New, less toxic drugs are needed to treat drug-resistant tuberculosis. Healthcare providers’ knowledge and vigilance regarding adverse drug reactions are important.

## Figures and Tables

**Table 1 medicina-61-00911-t001:** Mechanism of drug action and adverse drug reaction, drug interactions, and risk factors for adverse drug reactions (additional information adopted from [4,5,6,70]).

Drug	Mechanism of Action	Mechanism of Adverse Drug Reaction	Drug Interactions	Risk Factors for Adverse Drug Reactions
First-line antituberculosis drugs				
Isoniazid	Diffuses into the bacterial cytoplasm and disrupts mycobacteria’s mycolic acid synthesis	Toxic metabolites cause liver damage. Disrupts the metabolism of pyridoxine.	Foods, antacids/aluminum hydroxide, corticosteroids (decreased absorption and/or serum levels of isoniazid) Valproic acid, oral anticoagulants, benzodiazepines, carbamazepine, diazepam, levodopa, phenytoin, and theophylline (increased serum levels of those drugs or their effectiveness) Enflurane (possibility of nephrotoxicity) Ketoconazole (decreased serum concentration of ketoconazole) Cycloserine (greater neurotoxicity) Disulfiram (possibility of psychotic events) Paracetamol, rifampin (greater hepatotoxicity)	Older age, daily alcohol consumption, pre-existing liver disease, hepatitis C *. Genetic slow acetylator phenotype. For peripheral neuropathy—diabetes, HIV infection, uremia, malnutrition, and alcoholism.
Rifampicin	Penetrates the cytoplasm and inhibits RNA synthesis	Liver injury due to hypersensitivity reactions. Hepatic porphyria due to protoporphyrin IX toxicity if used in combination with isoniazid.	Foods and para-aminosalicylic acid (decreased absorption of rifampin) Amiodarone, oral anticoagulants, contraceptives, anticonvulsants, tricyclic antidepressants, antipsychotics, barbiturates and benzodiazepines, beta-blockers, cyclosporine, ketoconazole, codeine, corticosteroids, dapsone, digitalis, diltiazem, enalapril, statins, fluconazole, haloperidol, oral hypoglycemic agents, itraconazole, methadone, morphine, narcotics and analgesics, propafenone, nifedipine, quinidine, theophylline, verapamil, efavirenz, indinavir, lopinavir/ritonavir, nelfinavir, saquinavir, zidovudine (decreased serum levels of those drugs or reduced their effectivity) Isoniazid + ketoconazole, ethionamide, phenytoin, isoniazid, sulfonamides (greater hepatotoxicity) Pyrazinamide (greater uric acid excretion)	Age over 60 years and HIV infection
Pyrazinamide	It damages the cellular wall by inhibiting fatty acid synthesis.	Toxic effect on the liver through metabolites. Inhibition of urine acid transporters in the kidneys.	Allopurinol, colchicine (decreased effect of these drugs; pyrazinamide increases the serum levels of uric acid) Cyclosporine (decreased serum concentration of cyclosporine) Ethionamide (the adverse effects of ethionamide can increase) Rifampin, isoniazid, ketoconazole (greater hepatotoxicity)	Older age, diabetes mellitus * Pre-existing liver or kidney disease, elevated uric acid levels.
Ethambutol	Competes for binding sites to the EmbB and EmbC subunits, disrupting cell wall biosynthesis.	Optic neuropathy mechanisms may involve mitochondrial toxicity and disruption of axonal transport in the optic nerve.	Antacids (decreased absorption of ethambutol) Ethionamide (increased possibility of neurotoxic effects of ethambutol) Pyrazinamide (increased possibility of hepatotoxicity) Didanosine and zalcitabine (peripheral neuritis is potentiated)	High doses of ethambutol, long treatment duration, renal impairment, older age, and pre-existing optic nerve disease.
			Second-line antituberculosis drugs	
Fluoroquinolones	Inhibits DNA gyrase, leaving its ends free. It results in uncontrollable synthesis of mRNAs, exonuclease, and some proteins. Chromosomes degrade.	Block cardiac potassium channels. May involve the production of reactive oxygen species and matrix metalloproteinase activation.	Antacids with cations Ca, Mg, Al, and Fe; sucralfate (decreased absorption of fluoroquinolones) Drugs metabolized by cytochrome P450: cyclosporine, theophylline, warfarin, phenytoin, and sulfonylurea (increased effect of these drugs) Nonsteroidal anti-inflammatory drugs (increased stimulation of the central nervous system and possibility of convulsions) Probenecid (increased serum levels of the fluoroquinolone) Theophylline (increased serum levels of theophylline)	QT prolongation: elderly, female sex, electrolyte imbalance, cardiac diseases, use of other agents that prolong the QTc interval. Tendon rupture: older age, renal insufficiency, corticosteroid therapy.
Linezolid	Disrupts protein synthesis by binding to rRNA and inhibits the initiation process for protein synthesis.	Disrupts mitochondrial function.	Not enough data about interactions with rifampicin. Serotonin reuptake inhibitors.	Thrombocytopenia: low baseline platelet count, minimum concentration, renal insufficiency.
Clofazimine	Increases reactive oxidant species and destabilizes the bacterial membrane. Reverses the inhibition of intracellular phagocyte killing mechanisms and acts synergistically with interferon-gamma.	The drug accumulates in subcutaneous fat. Inhibits hERG cardiac potassium channels.	No data on drug interactions found. Additive QT prolongation can occur when used together with: fluoroquinolones, bedaquiline, delamanid; azoles, macrolides, metoclopramide, efavirenz, furosemide, hydrochlorothiazide, citalopram, escitalopram, methadone, antiarrhythmics, and others.	No data.
Cycloserine/Terizidone	Inhibits the D-alanyl-D-alanine synthetase, alanine racemase, and alanine permease. These enzymes are involved in the bacterial cell membranes’ peptidoglycan formation.	Acts as a partial NMDA receptor agonist, which may contribute to neuropsychiatric effects.	Alcohol (increased effects of alcohol and dizziness) Anticoagulants (increased serum concentration of the anticoagulant) Ethionamide, isoniazid (possibility of increased toxic effects on the central nervous system) Phenytoin (increased serum concentration of phenytoin) Vitamin B6 (increased vitamin B6 clearance)	Higher doses, longer treatment duration
Carbapenems	Enters the bacterial cell and acylate the PBP enzymes, which catalyze peptidoglycan formation, causing autolysis and cell damage due to osmotic pressure. Meropenem acts as a β-lactamase inhibitor.	Can bind to GABA receptors. Disrupts gut flora.	Ganciclovir (elevated risk for convulsions) Imipenem (lower serum concentration of valproate)	History of seizures.
Amikacin	Bind to the bacterial surfaces, penetrates into the cytoplasm, and interferes with the translation of proteins, damaging the cytoplasmic membrane.	Damage to cranial nerve VIII. Accumulation of the drugs in renal tubules.	Acyclovir, amphotericin, cephalosporins, cisplatin, cyclosporine (increased possibility of nephrotoxicity) Ethacrynic acid (increased possibility of ototoxicity) Oral anticoagulants (greater effect of the anticoagulant) Nonsteroidal anti-inflammatory drugs (increased possibility of ototoxicity and nephrotoxicity) Capreomycin (increased possibility of ototoxicity and nephrotoxicity) Furosemide (increased possibility of ototoxicity) Methotrexate (possible increase in the toxicity of methotrexate) Polymyxins (greater nephrotoxicity) Vancomycin (greater ototoxicity and nephrotoxicity) Neuromuscular blocking agents (additive effect)	Age, long treatment duration, high total accumulated dose, concomitant usage of diuretics, dehydration, and history of hearing impairment.
Ethionamide/prothionamide	The primary target of ethionamide is InhA, resulting in inhibition of mycolic acid biosynthesis.	No data.	Alcohol (increased possibility of psychotic reactions) Antituberculosis drugs (greater adverse effects) Isoniazid (temporarily increased serum concentration of isoniazid) Para-aminosalicylic acid (increased possibility of hypothyroidism) Dapsone (peripheral neuritis is potentiated)	For hepatotoxicity, history of liver disease and alcoholism. If mental instability in history, administer ethionamide with caution.
Para-aminosalicylic acid	Acts as a competitive inhibitor of para-aminobenzoic acid in folate synthesis by targeting dihydropteroate synthase. This leads to inhibition of dihydrofolate reductase and impaired bacterial growth. Metabolites can inhibit thymidylate synthase, disrupting DNA synthesis.	Immune responses to the drug or its metabolites, thyroid hormone synthesis interference, and the drug’s impact on intestinal function (reduces vitamin B12 absorption).	Anticoagulants, sulfonylurea (possibility of increased drug effect) Digoxin, vitamin B12 (decreased serum levels of drug/vitamin) Corticosteroids (possibility of increased adverse effects of the corticosteroid) Ethionamide (increased possibility of hypothyroidism and hepatotoxicity) Isoniazid (possibility of increased serum levels of isoniazid) Probenecid (increased serum concentration of para-aminosalicylic acid) Sulfonylurea (possibility of increasing hypoglycemic effects of sulfonylurea)	No data.
Bedaquiline	Inhibits mycobacteria-specific F-ATP synthase by binding to the c subunit. This halts ATP production.	Inhibits cardiac hERG potassium ion channels. Mitochondrial dysfunction and alterations in cellular signaling pathways (hepatotoxicity).	Rifamycins (decreased serum levels of bedaquiline) Azole antifungals, macrolides (increased serum levels of bedaquiline) Efavirenz, phenytoin, glucocorticoids, metoclopramide, furosemide, hydrochlorothiazide, citalopram, escitalopram, methadone, antiarrhythmics, fluoroquinolones, clofazimine, and delamanid (these drugs may add risk for QTc prolongation)	No data.
Delamanid	Disrupts the mycobacterial cell wall by inhibiting the synthesis of methoxy-mycolic and keto-mycolic acids. During activation by the enzyme deazaflavin-dependent nitroreductase, delamanid produces reactive nitrogen species.	Probably converted to primary metabolite DM-6705 (toxic).	Fluoroquinolones, clofazimine, bedaquiline, macrolides, metoclopramide, efavirenz, furosemide, hydrochlorothiazide, citalopram, escitalopram, methadone, and antiarrhythmics (these drugs prolong QTc themselves) Rifamycin, carbamazepine, ritonavir, ketoconazole Cycloserine/terizidone (higher risk for neuropsychiatric symptoms)	Hypoalbuminemia *
Pretomanid	In an aerobic setting, it inhibits protein and lipid synthesis, decreasing the availability of mycolic acids; in an anaerobic state, it generates des-nitro metabolites and releases nitric oxide, reducing adenosine triphosphate concentration in cells.	Animal studies show hepatic, ophthalmologic, and reproductive organ damage.	Rifampicin, efavirenz, and other strong CYP3A4 inducers (significantly decreased pretomanid serum concentration) Lopinavir/ritonavir, other mild CYP3A4 inducers (smaller effect)	No data.

* Data varies across studies.

**Table 2 medicina-61-00911-t002:** Timing, frequency, monitoring, and management, discontinuation rates due to adverse drug reactions (additional information adopted from [4,5,6,70]).

Drug	Timing of Adverse Drug Reactions	Common/Rare Adverse Drug Reactions	Monitoring for Adverse Drug Reactions	Management of Adverse Drug Reactions	Discontinuation Due to Adverse Drug Reaction Rate
Isoniazid	Hepatotoxicity—first few weeks. Peripheral neuropathy—gradually after several weeks or months.	*Common* Elevated liver enzymes and gastrointestinal disturbances. *Less common, more severe* Peripheral neuropathy, hepatotoxicity. *Rare* Severe skin reactions, psychosis, depression, dysphoria, irritability, seizures, optic neuritis, dysarthria, pancreatitis, vasculitis, arthralgia, anemia, and thrombocytopenia.	Mental health/neuropsychiatric assessment, HIV, ALT, creatinine, complete blood count, hepatitis B/C serology, and glycated hemoglobin. Laboratory test monitoring without symptoms or baseline abnormalities may not be needed unless risk factors are present. Rash, if severe, assess for organ dysfunction: LFT/creatinine, eosinophils (DRESS syndrome).	Pyridoxine supplementation. Hypersensitivity skin reactions—antihistamines, corticosteroids, and, in severe cases, desensitization. Psychiatric disorders—administer psychiatric medication; in severe cases, discontinue the drug.	0.99%—hepatotoxicity, 0.89%—fever, 0.79%—peripheral neuropathy, 0.69%—skin reactions, 0.49%—hypersensitivity, and 0.18%—psychiatric changes.
Rifampicin	Gastrointestinal disturbances—during the first month.	*Common* Rash, gastrointestinal disturbances, and elevated liver enzymes. *Less common, more severe* Hepatotoxicity, thrombocytopenia, acute renal failure, and flu-like syndrome. *Rare* Hemolytic anemia, pseudomembranous colitis, pseudoadrenal crisis.	HIV, ALT, creatinine, complete blood count, hepatitis B/C serology, and glycated hemoglobin. Laboratory test monitoring without symptoms or baseline abnormalities may not be needed unless risk factors are present. Rash, if severe, assess for organ dysfunction: LFT/creatinine, eosinophils (DRESS syndrome).	Liver enzyme monitoring. Gastrointestinal disturbances can be managed symptomatically. If thrombocytopenia occurs, discontinue the drug, corticosteroids, and platelet transfusions in some cases.	0.6–1.19%
Pyrazinamide	Hepatotoxicity and hyperuricemia—most common during the first 2–9 weeks.	*Common* Hyperuricemia, arthralgia, elevated liver enzymes, exanthema, and gastrointestinal disorders. *Less common, more severe* Hepatotoxicity, gout, rhabdomyolysis. *Rare* Toxic epidermal necrolysis, photosensitivity, thrombocytopenia, and sideroblastic anemia.	Liver function (AST, ALT, and bilirubin) should be monitored at baseline and monthly if possible. Patients should be closely monitored if they are at risk for drug-related hepatitis and if signs or symptoms of hepatotoxicity occur.	If symptomatic hyperuricemia, allopurinol should be administered. Arthralgia can be managed symptomatically.	0.6–5% In elderly patients, 20.6%.
Ethambutol	Optic neuropathy develops after several weeks or months.	*Common* Optic neuropathy. *Rare* Peripheral neuropathy, rash, gastrointestinal disturbances, thrombocytopenia, cutaneous reaction, and acute renal failure	Patients should report any vision changes. Baseline and monthly visual acuity and color discrimination monitoring should be performed; high risk for patients on higher doses or with renal impairment. Each eye must be tested separately, and both eyes tested together.	An ophthalmologist consultation is needed. Visual acuity and fields should be monitored. Immediately discontinue if visual symptoms occur.	0.2–1% at 15 mg/kg dosage. 15–18% at 35 mg/kg dosage.
Fluoroquinolones	Adverse drug reactions to moxifloxacin in a median of 15 days and to levofloxacin in a median of 35 days. Tendon rupture—early during treatment and weeks to months after.	*Common* Gastrointestinal disturbances, nausea, diarrhea, dizziness, insomnia, and skin rash. *Less common, more severe* QT prolongation, arrhythmia, tendon rupture, peripheral neuropathy, hallucinations, delusions, pseudomembranous colitis, urticaria, vasculitis	Symptomatic monitoring. ECG should be carried out before treatment and at least 2, 12, and 24 weeks after starting treatment. More frequent monitoring is required if cardiac conditions, hypothyroidism, or electrolyte disturbances are present.	Correct electrolyte imbalances. Fluoroquinolones should be stopped if the QTc > 500 msec, and ECGs and potassium should be monitored frequently until the QTc returns to normal.	Levofloxacin—3.2%, moxifloxacin—4.5%.
Linezolid	Myelosuppression occurs after 2–4 weeks of treatment, peripheral neuropathy—gradually after several weeks or months, and optic neuropathy—after more than 2 months of treatment.	*Common* Gastrointestinal disorders, peripheral neuropathy, and anemia. *Rare* Optic neuritis, thrombocytopenia, and lactic acidosis.	Monitor for: -peripheral neuropathy and optic neuritis, through visual eye acuity (both eyes) and Ishihara tests every 2 months or, if symptoms develop, clinical examination for peripheral neuropathy monthly; -complete blood count weekly during the initial period, then monthly, and thereafter as needed based on symptoms; -pH, anion gap, and lactate levels in case of suspected lactic acidosis (hyperlactatemia, if lactate >2.0 mmol/L and confirmed lactic acidosis at >4.0 mmol/L), hypotension, lethargy, or clinical worsening without a clear explanation.	If peripheral neuropathy or optic neuritis occurs, the drug should be discontinued. Anemia and thrombocytopenia could be reversed with dosage reduction.	14.1%
Clofazimine	Skin discoloration develops within a few weeks.	*Common* Skin discoloration, gastrointestinal disorders, skin dryness, and ichthyosis, QT prolongation *Rare* Hepatitis, hypersensitivity reaction, nephrotoxicity, and acne.	Monitor clinical signs and symptoms. Perform an ECG if other QT interval-prolonging agents are given concomitantly.	Skin discoloration—patient advisement. Gastrointestinal disorders may be managed symptomatically. QT prolongation—analysis and correction of electrolytes, cardiac diseases, and, if severe, discontinuation of the drug.	
Cycloserine/Terizidone	A median of 71 days.	*Common* Headache, tremors, sleep disturbances, anxiety, depression, confusion, pale skin. *Rare* Visual changes, skin rash, hepatitis, tingling, and numbness in the extremities.	Baseline and monthly monitoring for depression should be carried out using a tool (e.g., the Beck Depression Index). If therapeutic drug monitoring is possible, obtain peak concentrations within the first 1–2 weeks of therapy and monitor them serially during therapy, keeping peak concentrations at <35 mcg/mL. When administering delamanid and cycloserine concurrently, monitor for neuropsychiatric side effects.	If seizures or psychotic symptoms occur, discontinue the drug. Pyridoxine supplementation is recommended.	6.25–66%
Carbapenems	No data.	*Common* Gastrointestinal disturbances. *Less common* Pseudomembranous colitis. *Rare* Seizures (more linked to imipenem). Injection site inflammation.	Monitor clinical signs and symptoms.	Gastrointestinal symptoms—symptomatic management. If diarrhea and fever, test for pseudomembranous colitis.	4.9%
Amikacin		*Common* Pain at the injection site and proteinuria *Less common, more severe* Cochlear, vestibular, nephrotoxicity, peripheral neuropathy, rash, and eosinophilia.	• creatinine at least monthly (more frequently if there is renal or hepatic impairment). • Creatinine clearance if there is baseline renal impairment or any concerns. • Electrolytes: baseline follow-up with monthly minimum potassium, magnesium, and calcium if possible. Audiology examination: baseline and monthly. Vestibular examinations: Question patients regularly about symptoms and perform serial vestibular exams. If possible, in patients aged over 60 years or with altered renal function, peak serum concentrations should be monitored.	If hearing loss occurs, consider discontinuing the drug. If nephrotoxicity occurs, discontinue the drug. Manage electrolyte imbalance.	13.8%
Ethionamide/prothionamide	Hepatic adverse reactions can occur for up to five months after initial treatment.	*Common and severe* Gastrointestinal disturbances. *Less common* Hepatotoxicity, neurological and psychiatric disturbances, hypothyroidism, menstrual irregularity, gynecomastia, arthralgia, and leukopenia. *Rare* Peripheral and optic neuritis, rash, photosensitivity, and thrombocytopenia.	TSH should be monitored for evidence of hypothyroidism requiring replacement therapy. Therapeutic drug monitoring is required if malabsorption is suspected. Liver function tests should be monitored.	Pyridoxine supplementation is recommended. Gastrointestinal disturbances must be managed symptomatically. If psychiatric symptoms, consider specialist consultation. The drug may be stopped, and psychiatric drugs prescribed.	No data.
Para-aminosalicylic acid	Gastrointestinal intolerance occurs after one week of treatment or more.	*Common* Gastrointestinal disturbances, hypothyroidism. *Rare* Hepatitis, allergic reactions, hemolytic anemia, granulocytopenia, polyneuritis, pericarditis, and malabsorption.	Should monitor TSH, electrolytes, blood counts, and liver function tests.	Diarrhea may improve after several weeks of treatment, and nausea and vomiting can be managed symptomatically. Thyroid function normalizes after drug discontinuation; thyroxine therapy may be needed. Splitting the dose or timing with food sometimes alleviates symptoms.	11.6%
Bedaquiline	Most significant QTc prolongation—in 6 or 15 weeks. * 34.2% of reactions were in the first month, and 14.2% were in the second.	*Common* Nausea, vomiting, abdominal pain, anorexia, arthralgia, headache, and QTc prolongation. *Rare* Hyperuricemia, phospholipidosis, elevated transaminases, and pancreatitis.	ECG before treatment and at least 2, 12, and 24 weeks after starting treatment. More frequent monitoring is recommended if cardiac conditions, hypothyroidism, or electrolyte disturbances are present. Liver function tests should be conducted at baseline, then monthly.	Gastrointestinal symptoms and arthralgia can be managed symptomatically and improve after a few weeks of treatment. Bedaquiline should be stopped if the QTc > 500 msec, and ECGs and potassium levels should be monitored regularly until the QTc returns to normal.	1.7%
Delamanid	Changes in ECGs peak at week 8.	*Occasional* QTc prolongation, nausea, vomiting, dizziness, insomnia, anxiety, hallucinations, night terrors, and upper abdominal pain	Before treatment, ensure the albumin level is 2.8 g/dL or higher. ECG and baseline electrolytes should be obtained and repeated if necessary (e.g., documented QTc prolongation or multiple risk factors). When administering delamanid and cycloserine concurrently, monitor for neuropsychiatric side effects.	Gastrointestinal symptoms—symptomatic management.	2.5–3.8%
Pretomanid	No data.	*Common* Nausea and vomiting, acne, headache, musculoskeletal pain, and liver enzyme elevation.	Symptoms of hepatotoxicity should be monitored, and liver function tests should be performed at baseline, at 2 weeks, and then monthly as needed. ECG and baseline electrolytes should be obtained before the initiation of treatment and repeated if needed (e.g., documented QTc prolongation or multiple risk factors).	No data.	No data.

* Data varies across studies.

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
