# Peer review of "The Adverse Effects of Tuberculosis Treatment: A Comprehensive Literature Review"

_medicina, 2025, doi:10.3390/medicina61050911_

Round 1

Reviewer 1 Report

Comments and Suggestions for Authors

A review paper is an overview of current knowledge and identifying gaps rather than presenting original findings.

This review article is unique and too complete. However need to identify the gaps between presenting original findings and real situation.

This review paper looks like a text book picture.

Comments on the Quality of English Language

Most of the sentences are too long. Shorter one is better.

Author Response

Comments 1:  A review paper is an overview of current knowledge and identifying gaps rather than presenting original findings.

This review article is unique and too complete. However need to identify the gaps between presenting original findings and real situation.

This review paper looks like a text book picture.

Response 1:  We are very grateful for the effort you dedicated to reviewing our submission and your favorable comments. While doing research for this data, we noticed dissimilarities in numbers that should be parallel. In our opinion, these discrepancies are primarily due to the wide diversity of studies and analyses, where different terms and statistical methods are used. It is likely that different patient profiles are of great importance because we see large differences in studies conducted in different countries and settings. Based on our experience in a tertiary-level 80-bed tuberculosis inpatient unit, we believe that the reported frequency of adverse reactions and the associated risk factors do not always align with today’s reality. Since we don’t want to make assumptions and speculate about the numbers, we have recently started a clinical study on this topic.

We are encouraged by your comment: “The review paper looks like a text book picture.”.

Comments 2: Most of the sentences are too long. Shorter one is better.

Response 2: Thank you very much for your advice. We restructured some of the sentences.

Reviewer 2 Report

Comments and Suggestions for Authors

Rūta et al. comprehensively described the side effects of currently used drug regimens against tuberculosis. The review is well summarized and drafted.

It could be beyond the scope of this article to include the precautions and recommendations of the treatment of AMR tuberculosis and how to avoid the development of AMR.

Author Response

Comments 1: Rūta et al. comprehensively described the side effects of currently used drug regimens against tuberculosis. The review is well summarized and drafted.

It could be beyond the scope of this article to include the precautions and recommendations of the treatment of AMR tuberculosis and how to avoid the development of AMR.

Response 1: Thank you for your review. Your favorable comment motivates us to continue the work.

We did not fully understand the second part of this comment. Therefore, we have not yet been able to reply to it and revise the manuscript accordingly. Please kindly extract the abbreviation AMR.

Reviewer 3 Report

Comments and Suggestions for Authors

Given the essentiality of combination-based regimens for the treatment of tuberculosis (TB) as well as the long treatment durations of such regimens, drug-associated toxicity is a common feature of TB therapy. These toxicity problems have resulted to treatment discontinuation and poor adherence, indirectly contributing to the continuous evolution of drug-resistant TB. In this review, the authors tried to summarize some of the adverse effects that has been reported for some TB drugs. While the efforts of the authors are laudable, the review has serious weaknesses that need to be addressed by the authors. Here are few of them:

  1. The abstract is written more like an introduction, instead of a typical abstract. In the abstract, the authors need to basically summarize what the reader will see in the manuscript if the prospective reader is contemplating on reading the paper.
  2. The introduction needs significant re-writing to include pertinent information. For instance, there are other reviews that have been written for TB drug toxicology. The authors need to include these reviews and point out how their current review is different from already existing ones. They also need to include what they meant by first-line and second-line TB drugs in the introduction. Throughout the manuscript, the authors kept on using the term 'tuberculosis strains'. That's wrong. It should be Mycobacterium tuberculosis strains, and the authors should have used the introduction to lay the foundation on this etiologic agent before moving ahead with their proposed review.
  3. It is surprising that the authors are trying to review the toxicology of different TB drugs without mentioning the specific cytochrome P450 enzymes that are known to metabolize these drugs. This is essential since in some cases, the metabolites of the drug are the drivers of the adverse effects.
  4. Reading throughout the individual sections of the paper, the authors made little efforts to include the molecular basis for the toxic effects observed with some TB drugs. For example, how does isoniazid disrupt the metabolism of vitamin B6 and why does that lead to peripheral neuropathy? I am only using isoniazid as an example here, but that is applicable to other drugs mentioned in this paper. Even if the mechanism of toxicology is not known, the authors should make efforts to speculate what they think might be responsible -- while clearly pointing out that they are only speculating. 
  5. The authors got the antimycobacterial mechanism of action of some drugs wrong, or where non-specific in their discussion of such. For example, the target of isoniazid is InhA. The target of bedaquiline is NOT 'adenosine triphosphatase synthase'.  It is ATP synthase. I see the same problems when the authors were discussing about clofazamine, ethionamide, and indeed many more drugs in the paper. Be clearer and specific on how these drugs work against the bacteria.
  6. Lastly, I will encourage the authors to run their manuscript through a plagiarism checker. I noticed that some paragraphs were collected word for word from https://iris.who.int/bitstream/handle/10665/365309/9789240065352-eng.pdf. There are other papers that were plagiarized, but I would want to believe that it was unintended. Please do well to correct. 

Author Response

Comments 1: Given the essentiality of combination-based regimens for the treatment of tuberculosis (TB) as well as the long treatment durations of such regimens, drug-associated toxicity is a common feature of TB therapy. These toxicity problems have resulted to treatment discontinuation and poor adherence, indirectly contributing to the continuous evolution of drug-resistant TB. In this review, the authors tried to summarize some of the adverse effects that has been reported for some TB drugs. While the efforts of the authors are laudable, the review has serious weaknesses that need to be addressed by the authors. Here are few of them:

  1. The abstract is written more like an introduction, instead of a typical abstract. In the abstract, the authors need to basically summarize what the reader will see in the manuscript if the prospective reader is contemplating on reading the paper.
  2. The introduction needs significant re-writing to include pertinent information. For instance, there are other reviews that have been written for TB drug toxicology. The authors need to include these reviews and point out how their current review is different from already existing ones. They also need to include what they meant by first-line and second-line TB drugs in the introduction. Throughout the manuscript, the authors kept on using the term 'tuberculosis strains'. That's wrong. It should be Mycobacterium tuberculosis strains, and the authors should have used the introduction to lay the foundation on this etiologic agent before moving ahead with their proposed review.
  3. It is surprising that the authors are trying to review the toxicology of different TB drugs without mentioning the specific cytochrome P450 enzymes that are known to metabolize these drugs. This is essential since in some cases, the metabolites of the drug are the drivers of the adverse effects.
  4. Reading throughout the individual sections of the paper, the authors made little efforts to include the molecular basis for the toxic effects observed with some TB drugs. For example, how does isoniazid disrupt the metabolism of vitamin B6 and why does that lead to peripheral neuropathy? I am only using isoniazid as an example here, but that is applicable to other drugs mentioned in this paper. Even if the mechanism of toxicology is not known, the authors should make efforts to speculate what they think might be responsible -- while clearly pointing out that they are only speculating. 
  5. The authors got the antimycobacterial mechanism of action of some drugs wrong, or where non-specific in their discussion of such. For example, the target of isoniazid is InhA. The target of bedaquiline is NOT 'adenosine triphosphatase synthase'.  It is ATP synthase. I see the same problems when the authors were discussing about clofazamine, ethionamide, and indeed many more drugs in the paper. Be clearer and specific on how these drugs work against the bacteria.
  6. Lastly, I will encourage the authors to run their manuscript through a plagiarism checker. I noticed that some paragraphs were collected word for word from https://iris.who.int/bitstream/handle/10665/365309/9789240065352-eng.pdf. There are other papers that were plagiarized, but I would want to believe that it was unintended. Please do well to correct.

Response 1: We are grateful for the time and energy you expended on our behalf. We sincerely appreciate all insightful comments, corrections, and suggestions that helped us to improve the quality of our manuscript and carry out a major revision of the paper. We found your feedback very constructive and hope you find these revisions rise to your expectations.

  1. We have rewritten the abstract per your suggestion.
  2. We expanded the introduction, adding the information and correcting the mistakes you mentioned in your comment.
  3. Thank you for this point. Cytochromes are referenced at several points within the article, especially in discussions concerning drug interactions. Since we aren’t focusing on biochemistry or molecular biology, we did not expand on this topic in our review.
  4. We want to clarify that our goal was not to conduct an in-depth analysis of the drugs' mechanisms of action. Our review is dedicated to a broad audience, most of whom are not scientists but practitioners. Also, we thank you for the suggestion, but we do not believe we have the right to speculate on possible mechanisms of action without having evidence.
  5. We corrected the error you pointed out in the comment. As we already mentioned, detailed analysis of the mechanism of action is not our goal, we apologize if we misled you on this matter. However, we checked the mechanisms of action of the drugs and adjusted them.
  6. We aim to ensure that no intentional plagiarism has occurred. However, due to the use of commonly accepted terminology, standard references, and extensive citation, some similarities with existing publications are inevitable.

Round 2

Reviewer 1 Report

Comments and Suggestions for Authors

Thanks for the revised version followed by comments and suggestions.

Comments on the Quality of English Language

English is improved and more clear.